# JAK2-Mediated Phosphorylation of Stress-Induced Phosphoprotein-1 (STIP1) in Human Cells

**DOI:** 10.3390/ijms23052420

**Published:** 2022-02-22

**Authors:** Angel Chao, Min-Jie Liao, Shun-Hua Chen, Yun-Shien Lee, Chi-Neu Tsai, Chiao-Yun Lin, Chia-Lung Tsai

**Affiliations:** 1Department of Obstetrics and Gynecology, Linkou Chang Gung Memorial Hospital, Chang Gung University, College of Medicine, Taoyuan 333, Taiwan; drangiechao@gmail.com (A.C.); minmin123@cgmh.org.tw (M.-J.L.); 2Gynecologic Cancer Research Center, Linkou Chang Gung Memorial Hospital, Taoyuan 333, Taiwan; 3Department of Obstetrics and Gynecology, Keelung Chang Gung Memorial Hospital, Keelung 204, Taiwan; 4School of Nursing, Fooyin University, Kaohsiung 831, Taiwan; meow.cat.15@gmail.com; 5Genomic Medicine Research Core Laboratory, Linkou Chang Gung Memorial Hospital, Taoyuan 333, Taiwan; bojack@mail.mcu.edu.tw; 6Department of Biotechnology, Ming Chuan University, Taoyuan 333, Taiwan; 7Graduate Institute of Clinical Medical Sciences, Chang-Gung University, Taoyuan 333, Taiwan; pink7@mail.cgu.edu.tw; 8Department of Surgery, New Taipei Municipal Tucheng Hospital, New Taipei City 236, Taiwan

**Keywords:** STIP1, JAK2/STAT3 signaling, phosphorylation, cancer

## Abstract

Stress-induced phosphoprotein-1 (STIP1)—a heat shock protein (HSP)70/HSP90 adaptor protein—is commonly overexpressed in malignant cells, where it controls proliferation via multiple signaling pathways, including JAK2/STAT3. We have previously shown that STIP1 stabilizes the protein tyrosine kinase JAK2 in cancer cells via HSP90 binding. In this study, we demonstrate that STIP1 may act as a substrate for JAK2 and that phosphorylation of tyrosine residues 134 and 152 promoted STIP1 protein stability, induced its nuclear-cytoplasmic shuttling, and promoted its secretion into the extracellular space. We also found that JAK2-mediated STIP1 phosphorylation enhanced cell viability and increased resistance to cisplatin-induced cell death. Conversely, interference STIP1 with JAK2 interaction—attained either through site-directed mutagenesis or the use of cell-penetrating peptides—decreased JAK2 protein levels, ultimately leading to cell death. On analyzing human ovarian cancer specimens, JAK2 and STIP1 expression levels were found to be positively correlated with each other. Collectively, these results indicate that JAK2-mediated phosphorylation of STIP-1 is critical for sustaining the JAK2/STAT3 signaling pathway in cancer cells.

## 1. Introduction

Stress-induced phosphoprotein 1 (STIP1, Gene ID:10963)—an adaptor protein of the HSP70/HSP90 chaperone heterocomplex [1,2]—contains a nuclear localization signal regulated by cell cycle kinase phosphorylation [3] and plays a critical role in cell proliferation [4]. Significant increases in the expression of STIP1 have been reported in several solid tumors, including hepatocellular carcinoma [5], pancreatic cancer [6], ovarian cancer [7,8], colon cancer [9], breast cancer [10], and cholangiocellular carcinoma [11]. Conversely, targeted downregulation of endogenous STIP1 expression resulted in a decreased proliferation of pancreatic cancer, ovarian cancer, and osteosarcoma cells [12,13,14,15]. Interestingly, an increased STIP1 expression has emerged as a clinically useful biomarker that predicts nodal spread and unfavorable overall survival in patients with malignancies [16]. There is also evidence that STIP1 may act as a secretory protein to promote tumor progression. In this regard, increased STIP1 serum levels in patients with ovarian cancer and hepatocellular carcinoma have been shown to activate the ERK2, SMAD1/5, and PI3K-AKT signaling pathways [8,17,18].

The Janus kinase (JAK)/signal transducer and activator of transcription (STAT) pathway plays an essential role in tumorigenesis, cancer invasiveness, and anticancer drug resistance [19,20,21,22]. Additionally, silencing of STIP1 disrupts the formation of HSP90 chaperone complexes and downregulates JAK2 and phospho-STAT3 protein expression, followed by degradation of unfolded JAK2 [23]. At the cellular level, STIP1 silencing inhibits the downstream effects of JAK2/STAT3 signaling—including proliferation, migration, and epithelial-mesenchymal transition [23,24,25,26].

We have previously shown that STIP1 interacts with JAK2 to maintain its protein stability [23]. Here, we further examined the reciprocal interactions between STIP1 and JAK2 with the goal of advancing our understanding of their functional role in cancer cells. STIP1 was found to act as a substrate for JAK2, and phosphorylation of tyrosine residues 134 and 152 promoted STIP1 protein stability, induced its nuclear-cytoplasmic shuttling, and promoted its secretion into the extracellular space. We also demonstrate that JAK2-mediated STIP1 phosphorylation enhanced cell viability and increased resistance to cisplatin-induced cell death. Conversely, blockading of STIP1 phosphorylation through site-directed mutagenesis or the use of cell-penetrating peptides decreased JAK2 protein levels, ultimately decreasing cell viability.

As JAK2-mediated phosphorylation of STIP-1 was found to be critical for sustaining the JAK2/STAT3 signaling pathway in malignant cells, blockading of STIP1 phosphorylation by JAK2 provides an avenue for developing novel potential anticancer strategies.

## 2. Result

### 2.1. JAK2 Phosphorylates STIP1 at Tyrosine 134 and 152

To assess whether the tyrosine kinase JAK2 is capable of phosphorylating STIP1, we initially searched for potential STIP1 tyrosine phospho-acceptor sites (http://www.hprd.org/PhosphoMotif_finder; accessed on 7 May 2018). Through this approach, we identified five candidate tyrosine residues (Figure 1a). On examining the phosphorylation status of STIP1 following shJAK2-mediated inhibition of endogenous JAK2 expression, immunoprecipitation experiments revealed that JAK2 inhibition decreased tyrosine phosphorylation of STIP1 compared to control conditions (Figure 1b). We subsequently sought to identify the target tyrosine residues for JAK2-mediated STIP1 phosphorylation. To this aim, recombinant wild-type STIP1 was incubated [17] with a constitutively activated JAK2 mutant (harboring the V617F variant) [27] in the presence of ATP. For comparison purposes, the constitutively activated JAK2 mutant was incubated with five different STIP1 isoforms in which the candidate tyrosine hosphor-acceptor sites were replaced by other amino acids through site-directed mutagenesis. Upon quantification of tyrosine-phosphorylated STIP1 levels (Figure 1c), the most prominent decreases were observed for the STIP1 Y134A, Y152A single mutant, and Y134/152A double mutant (Figure 1d). Collectively, these results indicate that phosphorylation of STIP1, mediated by JAK2, mainly occurs on tyrosine residues at position 134 and 152.

### 2.2. STIP1 Phosphorylation Is Essential in Maintaining the JAK2/STAT3 Signaling Pathway

STIP1 plays a critical role in mediating the JAK2/STAT3 signaling pathway [23]. To further explore the role of STIP1 phosphorylation in this process, we examined phospho-STAT3 Y705 levels in HEK293 cells overexpressing wild-type STIP1 versus three different tyrosine phospho-acceptor sites STIP1 mutants (Y134A, Y152A, and Y134/152A double mutant). Phospho-STAT3 Y705 levels were found to decrease in all of the cells that overexpressed STIP1 mutants, with the most pronounced decline being evident for the double mutant isoform (Figure 2a). Similar findings were observed when experiments were repeated using ovarian cancer SKOV3 cells (Figure 2b). Pull-down assays using specific antibodies revealed that—compared with wild-type STIP1—the STIP1 single or double mutant reduced the reciprocal interactions between JAK2 and HSP90 (Figure 2c–e). These results indicate that phosphorylation of STIP1, mediated by JAK2, on tyrosine residues at position 134 and 152 is essential in maintaining the JAK2/STAT3 signaling pathway.

### 2.3. JAK2-Mediated STIP1 Phosphorylation Maintains Its Protein Stability

We further investigated how phosphorylation of STIP1, mediated by JAK2, may affect its protein stability. To this aim, STIP1 degradation rates were examined in HEK293 cells overexpressing wild-type STIP1 versus the STIP1 double mutant (Y134/152A) in the presence of the protein synthesis inhibitor cycloheximide. The STIP1 double mutant underwent more rapid degradation than the wild-type STIP1 protein (Figure 3a). Because ubiquitination reflects the extent of protein degradation, wild-type STIP1 and the STIP1 double mutant were pulled down from HEK293 cells in the presence of the proteasome inhibitor MG132, followed by the use of an anti-ubiquitin antibody to probe the ubiquitination state. Levels of the ubiquitinated STIP1 double mutant were higher compared with those of wild-type STIP1 (Figure 3b). Moreover, JAK inhibitor I effectively blocked STIP1 expression in SKOV3 cells (Figure 3c). Collectively, these data suggest that phosphorylation of STIP1, mediated by JAK2, maintains its protein stability.

### 2.4. JAK2-Mediated STIP1 Phosphorylation Inhibits Cisplatin-Induced Cell Death

Cisplatin resistance in ovarian cancer cells has been associated with an increased phospho-STAT3 expression [28]; conversely, both JAK2 and STAT3 inhibitors can enhance cisplatin sensitivity [29]. Since phosphorylation of STIP1 by JAK2 can maintain STIP1 protein stability which involves the regulation of STAT3 phosphorylation, we therefore examined how phosphorylation of STIP1, mediated by JAK2, may affect cisplatin-induced cell death. We therefore examined how phosphorylation of STIP1, mediated by JAK2, may affect cisplatin-induced cell death. To this aim, we investigated the viability of HEK293 and SKOV3 cells that overexpressed wild-type STIP1 and the STIP1 double mutant following exposure to cisplatin. Cells expressing the STIP1 double mutant showed the highest sensitivity to cisplatin -induced cell death (Figure 4a,c). Similar findings (Figure 4b,d) were obtained when the cleaved poly (ADP-ribose) polymerase-1 was used as a marker of cell death [30]. Two cell-penetrating tyrosine phosphorylation inhibitory peptides—termed Peptide 134 and Peptide 152 for tyrosine 134 and 152, respectively (Figure 5a)—effectively decreased both JAK2 and STIP1 expression levels in HEK293 and SKOV3 cells (Figure 5b). Additionally, concomitant exposure to both peptides reduced cell viability (Figure 5c) and increased cisplatin-induced cell viability (Figure 5d). Collectively, these findings indicate that phosphorylation of STIP1 mediated by JAK2 enhances cell viability and increases resistance to cisplatin-induced cell death.

### 2.5. JAK2-Mediated STIP1 Phosphorylation Regulates Nuclear-Cytoplasmic Shuttling and Its Secretion in the Extracellular Space

STIP1 can undergo nuclear-cytoplasmic shuttling [31] and be secreted in the extracellular space, where it acts as an autocrine and paracrine molecule to promote cell proliferation [17]. We initially examined the localization of wild-type STIP1 versus the STIP1 double mutant in the presence of the nuclear export inhibitor leptomycin B in HAP1 STIP1^KO^ cells. Regardless of exposure to leptomycin B, a predominantly cytoplasmic localization was observed for wild-type STIP1 (Figure 6a). However, the STIP1 double mutant showed a clear nuclear localization in cells exposed to leptomycin B (Figure 6b). As the mechanisms underlying STIP1 secretion in the extracellular space remain unclear [8,18,32,33,34], we examined whether JAK2 may be involved in this process. While overexpression of the constitutively activated JAK2 mutant (harboring the V617F variant) was found to increase wild-type STIP1 protein levels in culture medium (lanes 2 and 5, Figure 6c), those of the STIP1 double mutant were reduced (lanes 3 and 6, Figure 6c) in HEK293 cells. Similar findings were observed in HAP1 cells where STIP1 expression was silenced through CRISPR gene editing (Figure 6d). Taken together, these results indicate that JAK2-mediated STIP1 phosphorylation regulates nuclear-cytoplasmic shuttling and its secretion in the extracellular space.

### 2.6. STIP1 and JAK2 Expression Levels Are Positively Correlated to Each Other in Human Serous Ovarian Cancer

Cell lysates from human serous ovarian cancer were subjected to western blot to measure endogenous JAK2 and STIP1 protein expression (Figure 7a). Expression levels of the two molecules were positively correlated to each other (r = 0.56, *p* < 0.05, Figure 7b)—a finding which was independently confirmed using the ovarian cancer RNA expression data from The Cancer Genome Atlas (r = 0.23, *p* < 0.01, Figure 7c, http://www.cbioportal.org/, accessed on 7 May 2018). These results suggest that JAK2 and STIP1 protein expression increase in parallel in human serous ovarian cancer.

## 3. Discussion

Our results indicate that JAK2 is capable of phosphorylating STIP1 on tyrosine residues at position 134 and 152—an event which plays a critical role in maintaining the JAK2/STAT3 signaling pathway. Importantly, JAK2-mediated STIP1 phosphorylation effectively inhibited cisplatin-induced cell death. In addition, JAK2 was identified as a key regulator of STIP1 nuclear-cytoplasmic shuttling and its secretion to the extracellular space (Figure 7d). Finally, expression of STIP1 and JAK2 were positively correlated to each other in human serous ovarian cancer specimens. This is, to our knowledge, the first study to demonstrate that tyrosine phosphorylation of STIP1, mediated by JAK2, can affect a number of biological functions in cancer cells.

JAK2/STAT3 signaling plays a crucial role in tumorigenesis—including cell proliferation and migration [35], radioresistance [36], and cancer cell stemness [37], and strategies targeting this pathway hold promise in anticancer drug development [38]. Cisplatin forms DNA adducts on DNA to induce DNA damage by preventing DNA repair, and finally triggers apoptosis [39]. The combined treatment of cisplatin and JAK2 inhibitors can enhance cisplatin-induced cell death [40,41,42]. Nevertheless, the actual mechanism in this combined therapy is still unclear. However, consecutive activated JAK2 can resist DNA-damage-induced apoptosis [43]. This may explain the efficacy of the combined therapy. We have previously shown that STIP1 maintains JAK2 protein stability and prevents apoptosis in ovarian cancer [23]. Interestingly, there have been reports showing that STIP1 can support the growth and spread of lung cancer [24] and melanoma [26] through the JAK2/STAT3 pathway, a finding in accordance with our current preclinical data. In this study, we expanded our previous results [23] by showing that JAK2 is critical for the stability and biological functions of STIP1; notably, the reciprocal interaction of these two molecules effectively maintained the JAK2/STAT3 signaling pathway. Since STIP1 protein is important for JAK2 protein stability, and JAK2 maintains STIP1 protein levels by phosphorylation, we proposed that inhibition of STIP1 phosphorylation may induce STIP1 degradation, repress JAK2 protein expression, and enhance therapeutic efficacy of cisplatin. As JAK2-mediated STIP1 phosphorylation effectively inhibited cisplatin-induced cell death, STIP1 blockade is a promising strategy for inhibiting the JAK2/STAT3 signaling in malignant cells and enhancing their sensitivity to chemotherapy drugs. However, the limitation of this study was that the mechanism of combined therapy cisplatin and JAK2 inhibitor remains to be elucidated.

Among the HSP90 family, only two main HSP90 isoforms—HSP90α and HSP90β—have STIP1 binding motif-MEVVD in the C terminal, and the interactions have been reported on the UniProt website (https://www.uniprot.org/uniprot; accessed on 7 February 2022). These two proteins are located mainly in the cytoplasm and are highly homologous, with 85% sequence identity [44]. However, there are several biochemical and functional differences between these two proteins [45]. The major biochemical difference is that the efficiency of protein dimerization is more rapid in HSP90α than in HSP90β. These two isoforms also play different functions in cell differentiation and embryonic development in various organisms [45]. However, the influence of STIP1 in these isoforms is still unclear.

Under physiological conditions, STIP1 is predominantly localized in the cytoplasm, where it is involved in the maintenance of the HSP90 chaperone machinery. However, in response to cellular stressors such as G1/S phase arrest, inhibition of cdc2 kinase, or irradiation-induced DNA damage, it can be translocated into the nucleus [1,31]. Previous studies have shown that both casein kinase II- and Cdk1-mediated STIP1 phosphorylation [1,46] are capable of promoting this translocation. Phosphorylation on murine STIP1 S189 and T198 by casein kinase II (CKII) or Cdk1 shuttles STIP1 between cytoplasm and nucleus [1,46]. In the current study, we extended our previous study and demonstrated that dephosphorylated STIP1 at Y134/Y152 is essential for maintaining the JAK2/STAT3 signaling pathway and interference of the STIP1-HSP90 interaction. Moreover, we found that a STIP1 double mutant elicited the nuclear localization of STIP1 in cells exposed to the nuclear export inhibitor leptomycin B (Figure 6b). Interestingly, nuclear JAK2 phosphorylates NF1, RUSH1, and Histone 3, which act as protein kinases to regulate gene expression [47] in response to cellular stressors [48]. Nuclear JAK2 also interacts with mismatch repair proteins MSH2 and MSH6 in the nucleus in response to reactive oxygen species (ROS) [48]. Further research is needed to investigate the biological roles of JAK2-STIP1 complexes in the nucleus.

Studies have shown STIP1 can be secreted from cells to act as an autocrine and paracrine molecule to promote cell proliferation [17,49,50]. Here, we demonstrate that JAK2-mediated STIP1 phosphorylation plays a significant regulatory role in this process. Our results point out that consecutive activated JAK2 overexpressed cell culture medium has higher expression levels of STIP1. Accordingly, overexpression of the constitutively activated JAK2 mutant (harboring the V617F variant) in HEK293 and HAP1 STIP1^KO^ cells were found to increase wild-type STIP1 protein levels in culture medium, which were conversely reduced when the STIP1 double mutant was used (Figure 6c,d). Interestingly, a previous study reported that the phosphorylation status of JAK2 can regulate HSP90 secretion [51]. Higher amount of HSP90 β is identified in conditioned medium when two phosphorylated sites, S225 and S255, on HSP90 β are mutated to alanine [51]. Our findings raise the interesting possibility that JAK2 inhibition may be a viable strategy to inhibit STIP1 release in the extracellular space.

In previous clinical trials, JAK inhibitors did not effectively inhibit tumor growth—especially when JAK2 mutations constitutively activated downstream signaling [52]. Since JAK2 acts as a client protein for the HSP90-STIP1 complex, targeting specific co-chaperones or chaperone complexes can improve the effectiveness of JAK inhibitors [53]. Once the molecular chaperone machinery is inhibited, JAK2 is unable to fold correctly and the JAK2/STAT3 signaling pathway is blocked. Consequently, the ability to inhibit STIP1 by RNA silencing [23,24,26] or using cell-penetrating tyrosine phosphorylation inhibitory peptides—as in the current study—has the potential to enhance the therapeutic potential of JAK2 inhibitors; future translational research should examine this hypothesis more rigorously.

## 4. Materials and Methods

### 4.1. Cell Culture and Materials

HEK293 (human embryonic kidney cells) and SKOV3 (human ovarian cancer cells) were obtained from the American Type Culture Collection (Manassas, VA, USA). Cells were grown in DMEM/F12 containing 10% fetal bovine serum and 1% penicillin and streptomycin at 37 °C in a humidified 5% CO_2_ atmosphere. The human HAP1 cell line cells knocked out for STIP1 by CRISPR/Cas9 (HAP1 STIP1^KO^ cell) were purchased from Horizon Discovery (Cambridge, UK) and maintained in Iscove’s Modified Dulbecco’s Medium containing 10% fetal bovine serum. In vitro experiments were performed using the following drugs or peptides: a proteasome inhibitor (MG132; Sigma-Aldrich, St. Louis, MO, USA; concentration: 5 μM); a JAK2 inhibitor (JAK inhibitor I; Millipore, Burlington, MA, USA; concentration: 0.1 and 1 μM); a protein synthesis inhibitor (cycloheximide; Sigma-Aldrich; concentration: 25 μg/mL); a chemotherapy drug (cisplatin; Fresenius Kabi Oncology Ltd., Bad Homburg, Germany; concentration: 20 μM); a nuclear export inhibitor (leptomycin B; Cell Signaling Technology, Danvers, MA, USA; concentration: 20 nM); a STIP1 tyrosine phosphorylation site (Tyr 134) inhibitory peptide (peptide 134; sequence: (D-arginine)_8_-NPFNMPNLYQKLESDPR; Leadgene Biomedical, Tainan City, Taiwan; concentration: 10 μM); and a STIP1 tyrosine phosphorylation site (Tyr 152) inhibitory peptide (peptide 152; sequence: (D-arginine)_8_-RTLLSDPTYRELIEQLR; Leadgene Biomedical; concentration: 10 μM).

### 4.2. Collection of Human Ovarian Cancer Specimens

After obtaining written informed consent, collection of serous ovarian cancer specimens from women who had undergone surgery was performed under appropriate Institutional Review Board approval (Linkou Chang Gung Memorial Hospital approval number: 101-4771B).

### 4.3. DNA Constructs

Halo-STIP1 mutants and the NTAP-JAK2 V617F construct were amplified from Halo-STIP1 and NTAP-JAK2 [23], respectively, using overlap extension polymerase chain reaction (PCR) with a Q5 site-directed mutagenesis kit (New England Biolabs, Ipswich, MA, USA). Short hairpin RNAs (shRNAs) for control (sh-Control) or JAK2 (sh-JAK2) were obtained from the National RNAi Core Facility at Academia Sinica (Taipei, Taiwan). The sequences of primers used to generate DNA expression vectors are shown in Table 1.

### 4.4. DNA Transfection

DNA transfection of HEK293 cells and STIP1 KO HAP1 cells was performed using the GenJet reagent (SignaGen Laboratories, Frederick, MD, USA; 2 μL for 1μg of DNA). SKOV3 cells were transfected with jetPRIME (Polyplus-transfection SA, Illkirch, France) according to the manufacturer’s instructions.

### 4.5. Cell Viability Assay

Cells were seeded into 96-well plates and either transfected with Halo-STIP1 constructs or treated with inhibitory peptides for 24 h. Subsequently, cells were exposed to 20 µM cisplatin for an additional 24 h. Cell viability was determined using the CCK-8 assay (Biotools, Taipei, Taiwan) according to the manufacturer’s protocol.

### 4.6. Western Blot

Cells were lysed in RIPA buffer (150 mM NaCl, 20 mM Tris-Cl pH 7.5, 1% Triton X-100, 1% NP40, 0.1% SDS, and 0.5% deoxycholate) containing protease and phosphatase inhibitors (Bionovas, Toronto, ON, Canada). Cellular proteins were subsequently separated by sodium dodecyl sulphate-polyacrylamide gel electrophoresis (SDS-PAGE) and transferred onto nitrocellulose membranes [23]. The following antibodies were used in western blot experiments: JAK2 (ab108596, Abcam, Cambridge, UK), STIP1(GTX103068, GeneTex, Hsinchu, Taiwan), Halo tag (G921A, Promega, Madison, WI, USA), calmodulin-binding peptide (CBP) tag for the pNTAP expression vector (sc-33000, Santa Cruz Biotechnology, Dallas, TX, USA), ubiquitin (#3936, Cell Signaling Technology, Danvers, MA, USA), phospho-STAT3 Y705 (2236-1,epitomics), STAT3 (ab32500, Abcam), HSP90 (#4877s, Cell Signaling Technology, Danvers, MA, USA), EGFP (GTX33910, GeneTex), cleaved PARP (#9546s, Cell Signaling Technology), actin (sc-47778, Santa Cruz Biotechnology), and GAPDH (sc-47724, Santa Cruz Biotechnology). The corresponding horseradish peroxidase-conjugated antibodies were obtained from Santa Cruz Biotechnology, whereas chemiluminescence reagents were from Millipore. The signal intensity of autoradiograms was quantified using the ImageJ software after normalization to the corresponding expression of actin or GAPDH.

### 4.7. Immunoprecipitation

Cells were lysed in a lysis buffer (20 mM Tris-Cl pH7.4, 25 mM NaCl, and 0.1% NP40) containing protease inhibitors. Cellular proteins were subjected to an overnight incubation with streptavidin beads (Invitrogen, Waltham, MA, USA; used for the pNTAP vector), Halo beads (Promega; used for the Halo tag protein), or protein A beads (used for the specific antibodies) at 4 °C under agitation. After three washes with a buffer solution (20 mM Tris-Cl pH 7.4 and 25 mM NaCl), pulled-down complexes were subjected to SDS-PAGE and detected with specific antibodies.

### 4.8. Confocal Microscopy

After overnight transfection in a glass cover slip, HAP1-STIP1 KO cells were fixed with 4% paraformaldehyde, immersed in ice cold acetone for 20 min at −20 °C, and incubated with blocking buffer (Thermo Fisher Scientific, Waltham, MA, USA) for 1 h at room temperature. Halo-STIP1 was detected by incubating cells with an antibody raised against the Halo tag, followed by exposure to the corresponding fluorescent antibody (Alex-fluor-488; Invitrogen). Finally, slides were examined under a Leica TCS SP2 confocal laser scanning microscope (Leica Microsystems GmbH, Wetzlar, Germany).

### 4.9. Statistical Analysis

A detailed description of statistical tests is provided in the figure legends. The correlation between JAK2 and STIP1 mRNA levels was examined on the cBioPortal website (http://www.cbioportal.org/, accessed on 7 February 2022) using the TCGA sequence dataset for ovarian cancer mRNA in correlation analysis with the Spearman method. All analyses were performed using GraphPad Prism, version 6.0 (GraphPad Inc., San Diego, CA, USA).

## 5. Conclusions

In summary, the results of this study demonstrate that JAK2 is capable of inducing tyrosine phosphorylation on specific STIP1 phospho-acceptor sites (tyrosine residues 134 and 152). As JAK2-mediated phosphorylation of STIP-1 was found to be critical for sustaining the JAK2/STAT3 signaling pathway in malignant cells, blockade of STIP1 phosphorylation by JAK2 provides an avenue for developing novel potential anticancer strategies.

## Figures and Tables

**Figure 1 ijms-23-02420-f001:**
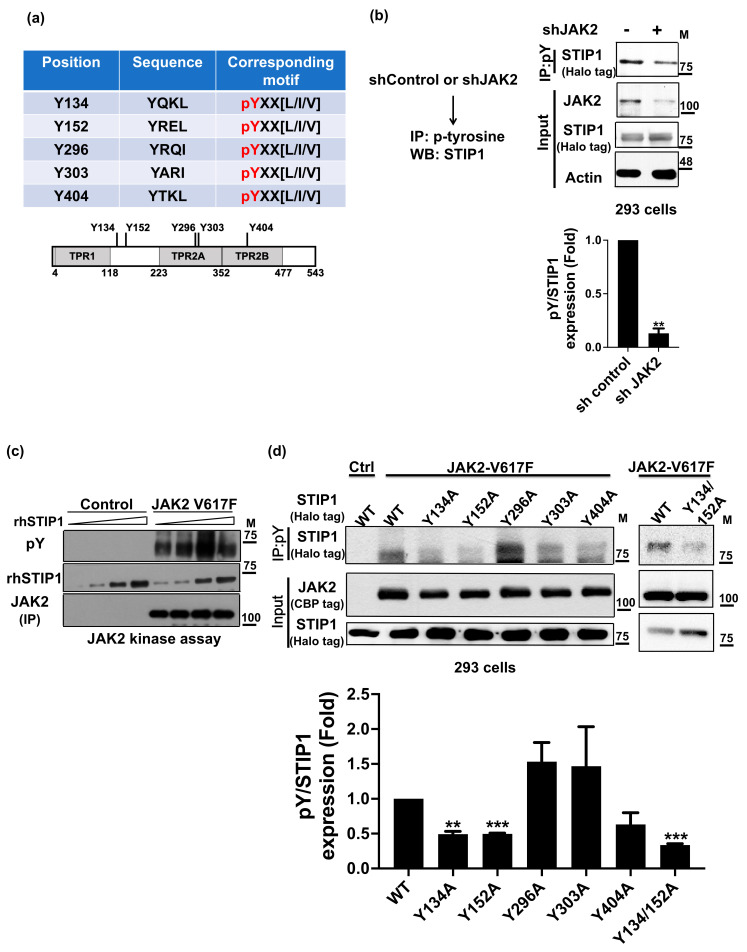
STIP1 undergoes JAK2-mediated phosphorylation. (**a**) Using the human protein reference database website (http://hprd.org/PhosphoMotif_finder; accessed on 7 May 2018), five potential tyrosine phospho-acceptor sites were identified within the STIP1 protein. (**b**) HEK293 cells underwent transfection with either sh-Control or sh-JAK2 for 48 h. Phosphorylated tyrosine proteins were purified with an anti-phospho-tyrosine antibody; the endogenous STIP1 protein was subsequently identified using a specific antibody. Fold changes in pulling-down phospho-STIP1 protein levels were reported in the bottom panel. Results are expressed as means ± standard errors of the mean from three independent experiments; ** *p* < 0.01 (**c**) In vitro kinase assay. A constitutively activated JAK2 mutant (harboring the V617F variant) purified from HEK293 cells with streptavidin beads was incubated with different doses of recombinant STIP1 (rhSTIP1) protein in presence of ATP. Phospho-STIP1 was detected using an anti-phospho-tyrosine antibody. In immunoprecipitation experiments, pulled-down rhSTIP1 and JAK2 (JAK2 IP) served as loading controls and were detected using specific antibodies. (**d**) Halo-tag of wild-type STIP1 or five single different tyrosine phospho-acceptor sites STIP1 mutants (Y134A, Y152A, Y296A, Y303A, and Y404A) and double mutant (Y134/152A) were co-transfected with NTAP-JAK2 V617F into HEK293 cells for 48 h. Proteins that were phosphorylated at tyrosine phospho-acceptor sites were immunoprecipitated with an anti-phospho-tyrosine antibody. The Halo-tag STIP1 proteins were identified using anti-STIP1 antibodies. The expression levels of NTAP-JAK2 (CBP tag) and Halo-STIP1 (Halo tag) in HEK293 cells are reported in the lower panel. Fold changes in phospho-STIP1 protein levels were reported in the bottom panel. Results are expressed as means ± standard errors of the mean from three independent experiments; ** *p* < 0.01, *** *p* < 0.001.

**Figure 2 ijms-23-02420-f002:**
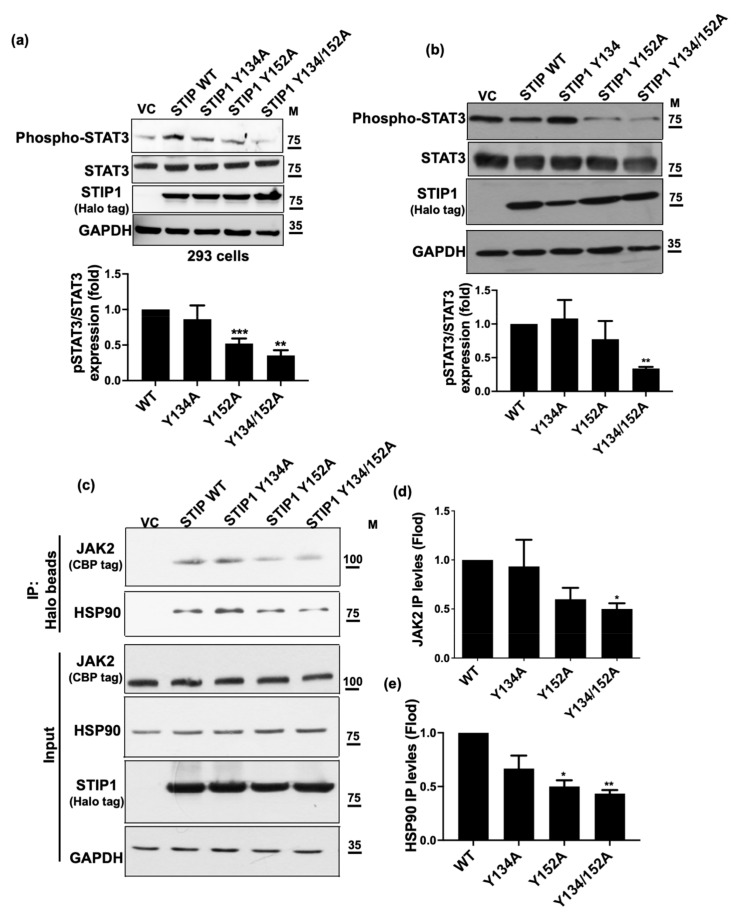
Phosphorylated STIP1 is essential for maintaining the JAK2/STAT3 signaling pathway. Halo-tag of wild-type STIP1 and three different tyrosine phospho-acceptor sites STIP1 mutants (Y134A, Y152A, and the Y134/152A double mutant) were transfected using expression vectors into (**a**) HEK293 cells and (**b**) SKOV3 cells for 48 h. Expression levels of endogenous phospho-STAT3 Y703, total-STAT3, GAPDH, and exogenous Halo-STIP1 were assessed with western blot using specific antibodies. GAPDH served as loading control. Fold changes in phospho-STAT3 protein levels were reported in the bottom panel. Results are expressed as means ± standard errors of the mean from three independent experiments; ** *p* < 0.01, *** *p* < 0.001. (**c**) Wild-type STIP1, STIP1 Y134A, STIP1 Y152A or the STIP1 Y134/152A double mutant (Halo tag) were co-transfected with JAK2 (CBP tag) and STAT3 (EGFP tag) into HEK293 cells. After pull-down with Halo resins (Halo-STIP1), purified JAK2 (CBP tag), STAT3 (EGFP tag), and endogenous HSP90 were detected using specific antibodies. (**d**) JAK2 and (**e**) HSP90 from (**c**) were quantified by the changes of protein levels. Results are expressed as mean ± standard errors from three independent experiments; * *p* < 0.05, ** *p* < 0.01.

**Figure 3 ijms-23-02420-f003:**
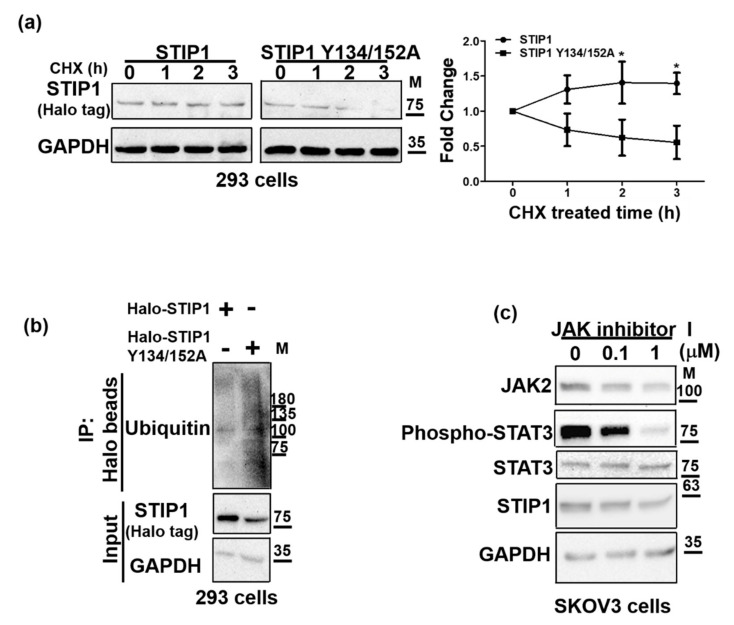
JAK2 regulates STIP1 stability. (**a**) HEK293 cells were transfected with wild-type STIP1 or the STIP1 Y134/152A double mutant (Halo tag) for 48 h and harvested at the reported time points in presence of cycloheximide (CHX). Endogenous GAPDH served as loading control to calculate normalized STIP1 protein levels. Fold changes in protein concentrations for Halo-STIP1 or Halo-STIP1 Y134/152A double mutant are reported in the right panel. Results are expressed as means ± standard errors from three independent experiments. * *p* < 0.05. (**b**) HEK293 cells overexpressing Halo-STIP1 or the Halo-STIP1 Y134/152A double mutant were harvested after treatment with MG132 (5 μM) for 12 h. Halo-STIP1 proteins were precipitated with a halo-resin, whereas ubiquitinated proteins were detected with an anti-ubiquitin antibody. Halo-STIP1 protein levels were detected with western blot using GAPDH as input control. (**c**) SKOV3 cells were treated with JAK inhibitor I in the DMEM/F12 medium containing 10% FBS for 48 h. Protein levels of JAK2, phospho-STAT3 Y705, total STAT3, STIP1, and GAPDH were assessed with western blot.

**Figure 4 ijms-23-02420-f004:**
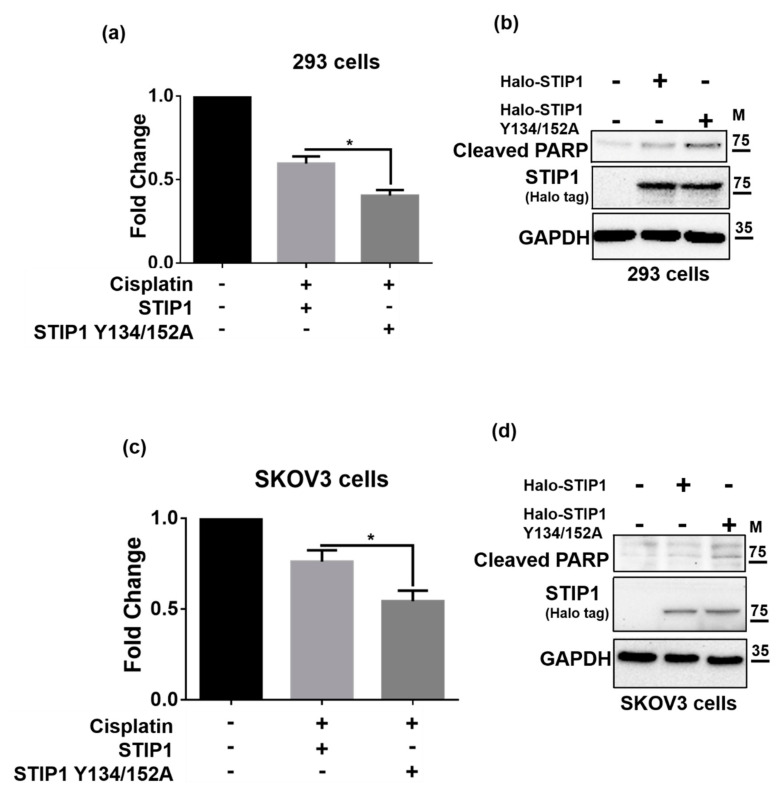
JAK2-mediated STIP1 phosphorylation inhibits cisplatin-induced cell death. (**a**,**b**) HEK293 cells and (**c**,**d**) SKOV3 cells were transfected with wild-type STIP1 or the STIP1 Y134/152A double mutant (Halo tag) for 24 h and subsequently treated with cisplatin (20 μM) for additional 24 h. (**a**,**c**) Cell viability was determined using the CCK-8 assay according to the manufacturer’s protocol. (**b**,**d**) Protein levels of cleaved PARP, exogenous STIP1 (Halo tag), and GAPDH were assessed with western blot. Results are expressed as mean ± standard errors from three independent experiments; * *p* < 0.05.

**Figure 5 ijms-23-02420-f005:**
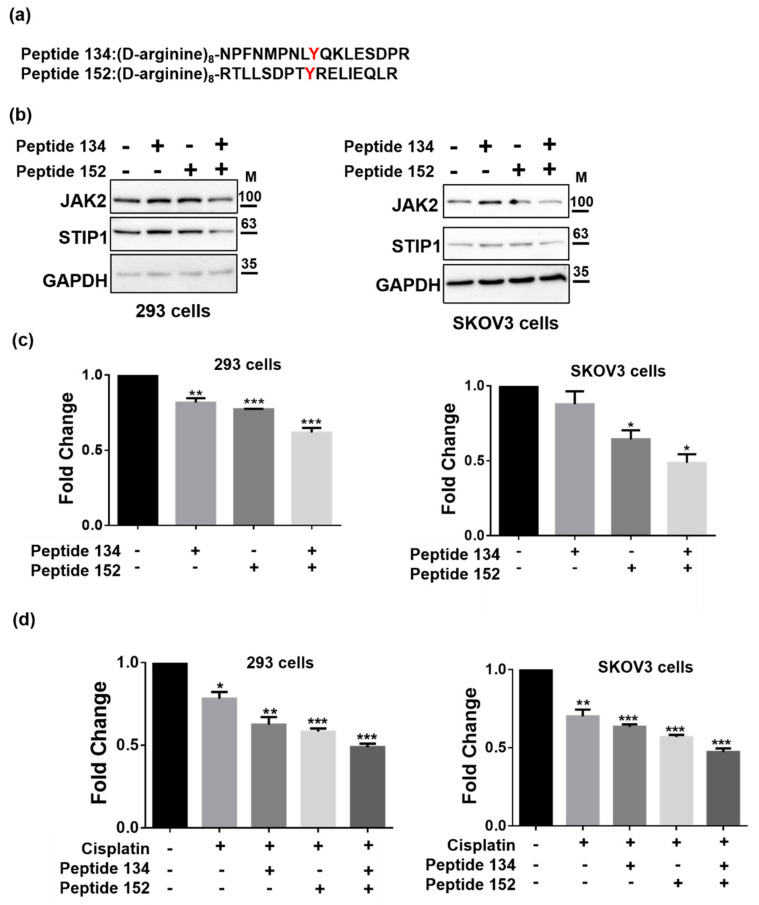
STIP1 Peptide 134 and Peptide 152 inhibit JAK2 and STIP1 expression. (**a**) Amino acid sequence of Peptide 134 and Peptide 152. Eight D-arginine residues were added to the N-terminal sequence of STIP1 peptides to serve as a cell membrane penetration signal. (**b**) HEK293 or SKOV3 cells were treated with Peptide 134 (10 μM), Peptide 152 (10 μM), or their combination (10 μM) for 48 h. JAK2, STIP1, and GAPDH protein levels were determined by western blot. (**c**) HEK293 and SKOV3 cells were treated with Peptide 134 (10 μM), Peptide 152 (10 μM), or their combination (10 μM) for 48 h. Cell viability was determined using the CCK-8 assay according to the manufacturer’s protocol. (**d**) HEK293 and SKOV3 cells were treated with Peptide 134 (10 μM), Peptide 152 (10 μM), or their combination (10 μM) for 24 h, and subsequently exposed to cisplatin (20 μM) for additional 24 h. Cell viability was determined using the CCK-8 assay according to the manufacturer’s protocol. Results are expressed as mean ± standard errors from three independent experiments; * *p* < 0.05, ** *p* < 0.01, *** *p* < 0.001.

**Figure 6 ijms-23-02420-f006:**
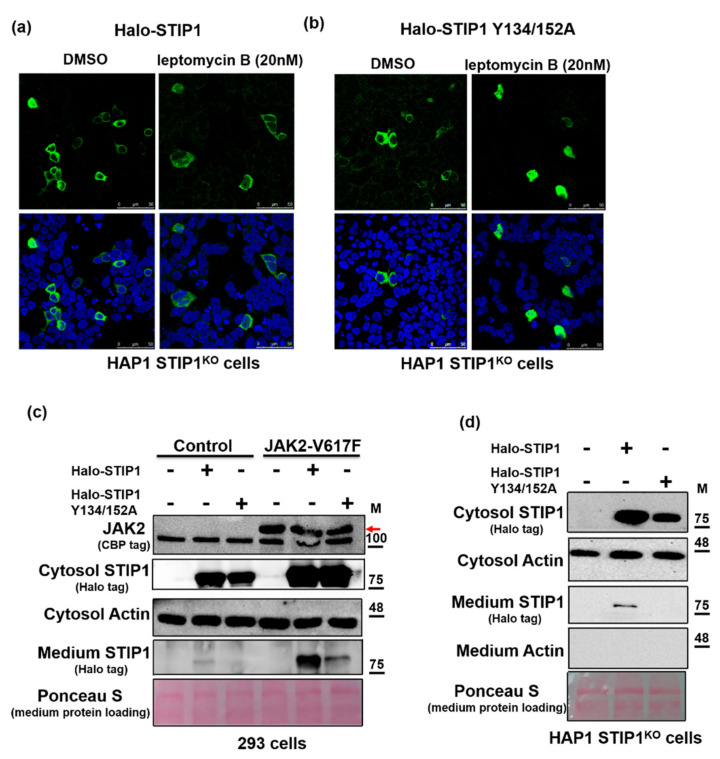
JAK2 regulates the intracellular localization and the extracellular secretion of STIP1. (**a**,**b**) Confocal microscopy was used to examine the intracellular localization of Halo-STIP1 (**a**) or Halo- STIP1 Y134/152A double mutant (**b**) in HAP1 STIP1 KO cells—either in presence or absence of leptomycin B (20 nM). Cells were probed with an anti-Halo tag antibody (green signal), whereas the nucleus was stained with DAPI (blue signal). (**c**) A control vector or the NTAP-JAK2 V617F vector were co-transfected with Halo-STIP1 or the Halo-STIP1 Y134/152A double mutant into HEK293 cells for 48 h. Protein levels of exogenous JAK2, exogenous STIP1, endogenous actin, and STIP1 in the culture medium were assessed with western blot. Ponceau S staining served as a loading control for STIP1 levels in the culture medium. The red arrow indicates the position of the JAK2 protein. (**d**) HAP1 STIP1 KO cells overexpressed wild-type STIP1 or the STIP1 Y134/152A double mutant for 48 h. Protein levels of exogenous STIP1, endogenous actin, and actin in the culture medium and STIP1 in the culture medium were assessed with western blot.

**Figure 7 ijms-23-02420-f007:**
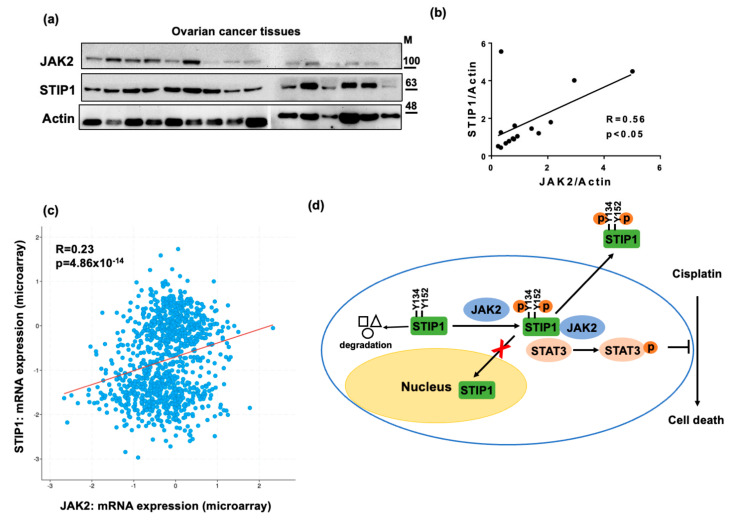
JAK2 and STIP1 protein expression levels are positively correlated to each other in human ovarian cancer tissue specimens. (**a**) JAK2 and STIP1 protein levels were determined by western blot in 15 serous ovarian cancer samples. Actin levels served as loading control. (**b**) Correlation between JAK2 and STIP1 protein levels in (**a**) after normalization to the corresponding actin intensity. (**c**) The correlation between JAK2 and STIP1 mRNA expression levels was investigated using data from Cancer Genome Atlas (TCGA) for ovarian cancer. (**d**) Key findings from this study: STIP1 phosphorylation—mediated by JAK2—promotes STIP1 stability, maintains the JAK2-STAT3 signaling pathway, is resistant to cisplatin treatment, and regulates STIP1 nuclear-cytoplasmic shuttling and its secretion in the extracellular space.

**Table 1 ijms-23-02420-t001:** DNA sequences of the primers used to generate constructs harboring point mutations.

Primer Name	Sequence (5′→3′)
JAK2 V617 F	TGGAGTATGTTTCTGTGGAGAC
JAK2 V617 R	TAATTTAAAACCAAATGCTTGTG
STIP1 Y134A F	GCCTAATCTGGCTCAGAAGTTGGAG
STIP1 Y134A R	ATGTTGAAAGGGTTCATGAATTTTC
STIP1 Y152A F	TGATCCTACCGCCCGGGAGCTG
STIP1 Y152A R	CTGAGTAGTGTCCTTGTC
STIP1 Y296A F	CCGAGAAGACGCTCGACAGATTGC
STIP1 Y296A R	TTTTCTCTCCCCACTTCAATG
STIP1 Y303A F	TGCCAAAGCAGCTGCTCGAATTG
STIP1 Y303A R	ATCTGTCGATAGTCTTCTC
STIP1 Y404A F	AGCTGCCTGCGCCACCAAACTC
STIP1 Y404A R	CGATTGCTGTATAATTTGGC

## Data Availability

The data will be available from the corresponding author upon request.

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
