# Peer review of "JAK2-Mediated Phosphorylation of Stress-Induced Phosphoprotein-1 (STIP1) in Human Cells"

_ijms, 2022, doi:10.3390/ijms23052420_

Round 1
Reviewer 1 Report
This manuscript describes the role of JAK2 in the phosphorylation of STIP1 and the impact on the efficacy of cisplatin’s inhibitory action on tumor cells. The interaction of JAK2 and STIP1 is well characterized using pull-down experiments and peptide inhibitors. However, the current description does not appropriately interpret the experimental result and the discussion section is not fully developed.
Major
- Y134A and Y152A: Figure 2 (a) included the analysis of the role of Y134A and Y152 independently and together, but Figure 2(b & c) only includes the combined effects. It is recommended to show that each of them is equally important and their effects are additive or synergistic.
- Hsp90: Hsp90 has multiple members and their functions may differ. It is recommended to show the specific isoform or all isoforms are involved in the response.
- Cisplatin: Although the alteration in the responses to cisplatin is shown, it is not clear how STIP1’s phosphorylation affects the action of cisplatin.
- positive correlation: The correlation coefficient of 0.25 is a weak correlation, especially in a log-log plot. The authors should check the positivity since the correlation coefficient may become large as long as the points are aligned on a line even the line is a vertical line with no positive/negative correlation.
- Figure 7d: This figure merely shows that STIP1 is phosphorylated by JAK2 and STIP1 (without phosphorylation) is transported to the extracellular domain. The illustration does not represent any of the important messages in this study. The authors should include the action of cisplatin, the role of STAT3 signaling, the involvement of Hsp90, etc.
- Discussion section: This section is missing and the description at the end of the results section is under-developed.
Minor
- Title: The role of JAK2 in STIP1’s phosphorylation is shown, but it is not clear whether this phosphorylation is critical for JAK2/STAT3 signaling. Little analysis on STAT3 is presented. Also, “cancer cells” is an overstatement since the cell lines in this study are not restricted.
- URL citation: Please check whether the citation of URL is appropriate in this journal.
Author Response
Reviewer 1
This manuscript describes the role of JAK2 in the phosphorylation of STIP1 and the impact on the efficacy of cisplatin’s inhibitory action on tumor cells. The interaction of JAK2 and STIP1 is well characterized using pull-down experiments and peptide inhibitors. However, the current description does not appropriately interpret the experimental result and the discussion section is not fully developed.
Major
- Y134A and Y152A: Figure 2 (a) included the analysis of the role of Y134A and Y152 independently and together, but Figure 2(b & c) only includes the combined effects. It is recommended to show that each of them is equally important and their effects are additive or synergistic.
A: Thank you for the suggestion. We initially tested the most important sites for JAK2 phosphorylation on STIP1. However, kinase assay data showed significant decrease in the phosphorylation levels of STIP1 in mutation of Y134 or Y152 STIP1 (Fig.1d). Double mutations at serine 134 and 152 showed the most prominent inhibitory activity against STAT3 phosphorylation (Fig.2a) and JAK2 protein stability (Fig. 5b) than single mutation, implying additive effects. According to these results, we believe these two sites are equally important to regulate STIP1 protein stability and maintain JAK2-STAT3 pathway and their effects are synergistic.
- Hsp90: Hsp90 has multiple members and their functions may differ. It is recommended to show the specific isoform or all isoforms are involved in the response.
A: We used general HSP90 antibody (mentioned in materials and methods) which detects endogenous levels of total HSP90 protein, alpha and beta isoforms in this study.
- Cisplatin: Although the alteration in the responses to cisplatin is shown, it is not clear how STIP1’s phosphorylation affects the action of cisplatin.
A: Because STIP1 phosphorylation is essential in maintaining the JAK2/STAT3 signaling pathway (Figure 2) and phospho-STAT3 participates in cisplatin resistance [28]. Therefore, STIP1’s phosphorylation may affect the action of cisplatin through STAT3 in this study. We have added “Since phosphorylation of STIP1 by JAK2 can maintain STIP1 protein stability which involves in the regulation of STAT3 phosphorylation, we therefore examined how phosphorylation of STIP1, mediated by JAK2, may affect cisplatin-induced cell death.” on lines 272-275, page 9.
- positive correlation: The correlation coefficient of 0.25 is a weak correlation, especially in a log-log plot. The authors should check the positivity since the correlation coefficient may become large as long as the points are aligned on a line even the line is a vertical line with no positive/negative correlation.
A: Percentage of tumor cells varies, may be as low as 30% in tissues causing RNA expression in complex tumor tissues may only have minor correlation. However, these two proteins have higher correlation in protein levels (Fig.7b, r = 0.56). Since JAK2 phosphorylated and regulated STIP1 in protein levels, we believe that correlations of protein levels better revealed the relations of these two proteins in tumor tissues.
- Figure 7d: This figure merely shows that STIP1 is phosphorylated by JAK2 and STIP1 (without phosphorylation) is transported to the extracellular domain. The illustration does not represent any of the important messages in this study. The authors should include the action of cisplatin, the role of STAT3 signaling, the involvement of Hsp90, etc.
A: Thank you for the suggestion. We have modified Fig.7d and on lines 360-361, page 13.
- Discussion section: This section is missing and the description at the end of the results section is under-developed.
A: Thank you for the suggestion. We have rewritten to discuss the biological functions of phosphorylated STIP1 which mediated by JAK2 on lines 390-414, page 14.
Minor
- Title: The role of JAK2 in STIP1’s phosphorylation is shown, but it is not clear whether this phosphorylation is critical for JAK2/STAT3 signaling. Little analysis on STAT3 is presented. Also, “cancer cells” is an overstatement since the cell lines in this study are not restricted.
A: The title has been changed to “JAK2-mediated phosphorylation of stress-induced phosphoprotein-1 (STIP1) in cancer cells”. Cancer cells is retained because 293 cells (human embryonic kidney cells) in this study is used as a “proof-of-principle” of this model.
- URL citation: Please check whether the citation of URL is appropriate in this journal.
A: Thank you. The citation of URL conforms to this journal.
Reviewer 2 Report
In this manuscript, Chao and colleagues present an in-depth study of the relationship between the kinase JAK2 and the protein STIP1, demonstrating STIP1 to be a substrate of JAK2 and that this has significant consequences for cell viability. The concept of the study is very strong, and the sequence of experiments performed is excellent. In principle, this would make a very good paper, however I have some significant experimental concerns, particularly in the early stages of the manuscript, which must be addressed.
Experiments in figures 1 and 2 must be quantified, to demonstrate that the apparent effect is consistent over multiple experiments.
In the case of the phosphorylation experiments in particular, it is also critical that the molecular weights of species are shown, so that the reader can confirm that the phospho-antibody signals represent the proteins expected. I suggest that the authors also supply complete blots as supplemental material.
In figure 1b, the STIP1(Halo tag) signal is extremely saturated. As a result, it is impossible to say whether the decrease in pY is due to reduced phosphorylation, or just lower levels of STIP1. As noted above, this must also be quantified.
In figure 1d, why was an additional experiment not done to see whether the Y134/Y152A double mutant showed lower phosphorylation that the equivalent single mutants? From figure 2, it is clear that the authors possess a plasmid that can express the double mutant. Additionally, in figure 1d, why is pY signal partially cut off?
Data on peptides 134 and 152 would be strengthened if the authors could also show a dose dependent effect of each peptide (separately), rather than just using one concentration.
Minor issues:
I would also include the abbreviation STIP1 in the title.
The level of detail in the methods is excellent, however can the authors please also give catalogue numbers for antibodies used, to improve repeatability.
It would be helpful if the borders of blots could be marked with a black line, particularly for those with very little background signal.
Author Response
In this manuscript, Chao and colleagues present an in-depth study of the relationship between the kinase JAK2 and the protein STIP1, demonstrating STIP1 to be a substrate of JAK2 and that this has significant consequences for cell viability. The concept of the study is very strong, and the sequence of experiments performed is excellent. In principle, this would make a very good paper, however I have some significant experimental concerns, particularly in the early stages of the manuscript, which must be addressed.
- Experiments in figures 1 and 2 must be quantified, to demonstrate that the apparent effect is consistent over multiple experiments.
A: Thank you for the suggestion. We have added quantified results in Figures 1 and 2 except Fig.1c, which we cannot measure the intensity since no bands in control group at pY level (pages 5 and 7).
- In the case of the phosphorylation experiments in particular, it is also critical that the molecular weights of species are shown, so that the reader can confirm that the phospho-antibody signals represent the proteins expected. I suggest that the authors also supply complete blots as supplemental material.
A: We have added the molecular weights markers in each western blot figures and also submitted un-processing western blot raw data. (pages 5, 7, 8, 10, 11, 12, 13 and supplementary data) (lines 193~195, 207~209, 222~223, 233~235, and 238~240).
- In figure 1b, the STIP1(Halo tag) signal is extremely saturated. As a result, it is impossible to say whether the decrease in pY is due to reduced phosphorylation, or just lower levels of STIP1. As noted above, this must also be quantified.
A: Thank you for the reminder. We have replaced the figure of halo-stip1 on page 6.
- In figure 1d, why was an additional experiment not done to see whether the Y134/Y152A double mutant showed lower phosphorylation that the equivalent single mutants? From figure 2, it is clear that the authors possess a plasmid that can express the double mutant. Additionally, in figure 1d, why is pY signal partially cut off?
A:
- In Figure 1d, we tried to find out which site is critical for JAK2 phosphorylation in Figure 1d. For this propose, we performed single mutant for kinase assay, instead of double mutant.
- In Figure 1d, we used anti-halo tag antibody to pull-down the Halo-STIP1, and the signal was very closed to the cut off IgG heavy chain (as shown below).
- Data on peptides 134 and 152 would be strengthened if the authors could also show a dose dependent effect of each peptide (separately), rather than just using one concentration.
A: Our previous study showed that the best treated concentration of STIP1was at 10 mM (reference#23 in this manuscript). Based on that experience, we used this concentration to perform experiments.
- Minor issues:
- I would also include the abbreviation STIP1 in the title.
A: Thank you. It is added.
- The level of detail in the methods is excellent, however can the authors please also give catalogue numbers for antibodies used, to improve repeatability.
A: We have added antibodies catalogue numbers on lines 131-139, page 4.
- It would be helpful if the borders of blots could be marked with a black line, particularly for those with very little background signal.
A: We have added the black borders in each western blot figures.
Round 2
Reviewer 1 Report
The responses by the authors are insufficient and the manuscript is not appropriate for publication.
Major
- Y134A and Y152A: The results are not clearly showing the role of each of Y134A and Y152A.
- Hsp90: Hsp90 has multiple members and their functions may differ. The analysis is insufficient.
- Cisplatin: The mechanism is not shown clearly, based on the specific action of cisplatin in DNA replication.
- positive correlation: The response is insufficient.
- Title: The analysis using a single cell line is inappropriate to state the results for cancer cells.
Author Response
- Y134A and Y152A: The results are not clearly showing the role of each of Y134A and Y152A.
A: Thank you for the suggestion. We have added Y134A and Y152A single mutations in Figures 2b and 2c. The results showed that each of them is equally important and their effects are additive when in combination of Y134A and Y152A on page 8.
- Hsp90: Hsp90 has multiple members and their functions may differ. The analysis is insufficient.
A: HSP90 can be classified into four groups in topology (as shown below; reference #28 of the revised manuscript). There were only two main HSP90 isoforms- HSP90a and HSP90b that have STIP1 binding motif-MEVVD in C terminal and the interactions have been reported in uniprot website (https://www.uniprot.org/uniprot). These two proteins major locate in cytoplasm and are highly homologous, with 85% sequence identity. Most researchers focus on STIP1 or JAK2/HSP90 do not distinguish them due to their similar structure and functions. In our previous study, we demonstrated that JAK2, STIP1, and HSP90 (α and β)) form a scaffold complex required for the transduction of JAK2-STAT3 signaling (reference #23 of the revised manuscript). In this study, we extended previous study and demonstrated that dephosphorylated STIP1 at Y134/Y152 is essential for maintaining the JAK2/STAT3 signaling pathway and interference of the STIP1-HSP90 interaction. We have added these sentences on lines 223-228, page 7; lines 409~411, page 15 and a new reference #28.
- Cisplatin: The mechanism is not shown clearly, based on the specific action of cisplatin in DNA replication.
A: The combined treatment of cisplatin and JAK2 inhibitors can enhance cisplatin-induced cell death and improve resistance (references #40-42 of the revised manuscript). However, the actual mechanism in this combined therapy is still unclear. Since STIP1 protein is important for JAK2 protein stability, and JAK2 can maintain STIP1 protein levels by phosphorylation, we proposed that inhibition of STIP1 phosphorylation may induce STIP1 degradation, repress JAK2 protein expression, and subsequently enhance the therapeutic efficacy of cisplatin. However, the limitation of this study was that the actual mechanism in cisplatin and JAK2 inhibitor combined therapy was not conducted. We have added these sentences on lines 383~384, 391~395, 398~400, page 15 and references 40-42 of the revised manuscript.
- positive correlation: The response is insufficient.
A: Percentage of tumor cells varies, may be as low as 30% of the tissues.
Therefore, RNA expression in complex tumor tissues may only have minor correlation. However, these two proteins have higher correlation in protein levels compared to RNA levels (Fig.7b, r = 0.56). Since JAK2 phosphorylated and regulated STIP1 in protein levels, we believe that correlations of protein levels better revealed the relations of these two proteins in tumor tissues. Similar situation was seen in breast cancer, for example, TGFß1 is one of LSD1 downstream genes in breast cancer (reference as shown below). However, the correlation of these two proteins in TCGA breast cancer database is -0.12 (http://gepia.cancer-pku.cn/index.html).
Reference
Wang, Y.; Zhang, H.; Chen, Y.; Sun, Y.; Yang, F.; Yu, W.; Liang, J.; Sun, L.; Yang, X.; Shi, L.; et al. LSD1 is a subunit of the NuRD complex and targets the metastasis programs in breast cancer. Cell 2009, 138, 660-672, doi:10.1016/j.cell.2009.05.050.
- Title: The analysis using a single cell line is inappropriate to state the results for cancer cells.
A: Thank you for the suggestion, we have replaced “cancer” into “human” in the title.

Reviewer 2 Report
The authors have addressed the majority of my concerns extremely well.
I would still like to see a blot in which their Y134/Y152A double mutant was stained with the pY antibody, compared to the WT and single mutant forms. This would solidify the argument that both sites are relevant, while also indicating whether additional sites may exist.
Once this is done, I recommend the manuscript for immediate publication.
Author Response
I would still like to see a blot in which their Y134/Y152A double mutant was stained with the pY antibody, compared to the WT and single mutant forms. This would solidify the argument that both sites are relevant, while also indicating whether additional sites may exist.
Once this is done, I recommend the manuscript for immediate publication.
A: Thank you for the suggestion. We have added Y134A/ Y152A double mutant in Figure 1d and modified the sentence on lines 184, page 5; lines 201-203, page 7.
Round 3
Reviewer 1 Report
The recommendation to the first revision was "reject" since the authors did not appropriately respond to the comments. Now the re-revised manuscript is improved. A further revision is recommended. Here are three comments to the re-revised version, which are almost the same as the first round of this review.
- Hsp90a and Hsp90beta: their functions can be different and it is recommended to conduct the experiment or state the potential difference.
- cisplatin: the authors need to consider the action of cisplatin in terms of DNA regulation and discuss the mechanism.
- correlation: even there is no functional link, the positive correlation can be strong if points, plotted on a plane, align close to a line with a positive slope. The authors are expected to show beyond the close alignment on a line.
Author Response
The recommendation to the first revision was "reject" since the authors did not appropriately respond to the comments. Now the re-revised manuscript is improved. A further revision is recommended. Here are three comments to the re-revised version, which are almost the same as the first round of this review.
- Hsp90a and Hsp90beta: their functions can be different and it is recommended to conduct the experiment or state the potential difference.
A: We have added the sentences” Among HSP90 family, only two main HSP90 isoform- HSP90a and HSP90b have STIP1 binding motif-MEVVD in C terminal and the interactions have been reported in uniprot website (https://www.uniprot.org/uniprot). These two proteins located in cytoplasm and are highly homologous, with 85% sequence identity (reference #44 of the revised manuscript). However, there are several biochemical and functional differences between these two proteins (reference #45 of the revised manuscript). The major biochemical difference is the efficiency of protein dimerization is more rapid in HSP90a than HSP90b. These two isoforms also play different functions in cell differentiation and embryonic development in various organisms (reference #45 of the revised manuscript). However, it is still unclear the influence of STIP1 in these isoforms.” on lines 400~409, page 15.
cisplatin: the authors need to consider the action of cisplatin in terms of DNA regulation and discuss the mechanism.
A: We have added the sentence “Cisplatin forms DNA adducts on DNA to induce DNA damage by preventing DNA repair, and finally triggers apoptosis (reference #39 of the revised manuscript). The combined treatment of cisplatin and JAK2 inhibitors can enhance cisplatin-induced cell death (reference #40~42 of the revised manuscript). Nevertheless, the actual mechanism in this combined therapy is still unclear. However, consecutive activated JAK2 can resist DNA damage induced apoptosis (reference #43 of the revised manuscript). It may explain the efficacy of the combined therapy.”, on lines 378~383, page 15.
correlation: even there is no functional link, the positive correlation can be strong if points, plotted on a plane, align close to a line with a positive slope. The authors are expected to show beyond the close alignment on a line.
A: In an attempt to show positive correlation between JAK2 and STIP1 mRNA, regression line of the two genes were plotted from BioPortal website (http://www.cbioportal.org/, Fig. 7c has thus been replaced in the revised manuscript.
